# UV photonic integrated circuits for far-field structured illumination autofluorescence microscopy

Chupao Lin [1,2] ✉, Juan Santo Domingo Peñaranda [3], Jolien Dendooven [3], Christophe Detavernier [3], David Schaubroeck [4], Nico Boon [5], Roel Baets [1,2] & Nicolas Le Thomas [1,2] ✉

Ultra-violet (UV) light has still a limited scope in optical microscopy despite its potential advantages over visible light in terms of optical resolution and of interaction with a wide variety of biological molecules. The main challenge is to control in a robust, compact and cost-effective way UV light beams at the level of a single optical spatial mode and concomitantly to minimize the light propagation loss. To tackle this challenge, we present here photonic integrated circuits made of aluminum oxide thin layers that are compatible with both UV light and high-volume manufacturing. These photonic circuits designed at a wavelength of 360 nm enable super-resolved structured illumination microscopy with conventional wide-field microscopes and without modifying the usual protocol for handling the object to be imaged. As a biological application, we show that our UV photonic chips enable to image the autofluorescence of yeast cells and reveal features unresolved with standard wide-field microscopy.

In the context of biological and chemical sensing, the UV wavelength range is particularly interesting as most of the biological and chemical molecules exhibit a strong absorption in this range. A large fraction of these molecules, among others the three amino acids, tryptophan, tyrosine and phenylalanine, and the nicotinamide adenine dinucleotide (NADH) coenzyme also provide fluorescence, which could be used advantageously for implementing label-free super-resolved microscopy techniques[1]. UV super-resolved microscopy techniques are still in their infancy, awaiting a technological breakthrough to allow their widespread use. One of the challenges facing standard UV bulk optics to achieve this goal is the lack of low-cost and aberration-free imaging lenses combining good transmission and high numerical aperture at UV and visibile wavelengths. Moreover, shaping and steering UV beams is crucial, in particular for structured illumination microscopy (SIM), which was originally implemented with bulky gratings[2,3] and more recently with spatial light modulators (SLM) or digital micro-

mirror devices (DMD), but only at wavelengths larger than 365 nm[4–6]. Combining mirrors and a SLM to control the phase and the orientation of the structured illumination has afterwards enabled to decouple the excitation and collection in order to enlarge the field of view up to 16 mm² with a collection numerical aperture of 0.25[7], but still in a bulky configuration and at visible wavelengths.

In this article, we unveil photonic integrated circuits (PICs) that are promising candidates for improved performance and widespread adoption of UV microscopy. PICs offer robust, compact and low-cost solutions for implementing complex optical functions that are hardly achievable at the same level of performance with standard bulk optics[8]. In particular, bulk optics cannot compete with the large-scale manufacturing capabilities offered by silicon photonics platforms[9,10]. PICs have already sparked a tremendous number of applications of paramount importance in telecommunications[11], healthcare[12], biological research[13], quantum information and computation[14,15], metrology[16,17],

[1]Photonics Research Group, INTEC Department, Ghent University-imec, 9052 Ghent, Belgium. [2]Center for Nano- and Biophotonics, Ghent University, Ghent, Belgium. [3]Department of Solid State Sciences, CoCooN, Ghent University, 9000 Ghent, Belgium. [4]Centre of Microsystems Technology (CMST), imec and Ghent University, 9052 Zwijnaarde, Belgium. [5]Center for Microbial Ecology and Technology (CMET), Ghent University, Gent, Belgium. ✉e-mail: Chupao.Lin@UGent.be; Nicolas.LeThomas@UGent.be

chemical[18,19] and biological sensing[20], environmental monitoring[21], and artificial intelligence[22]. With regard to super-resolution optical microscopy, PICs have been mainly used for near-field imaging, in particular for implementing wide field-of-view waveguide-based total internal reflection fluorescence (TIRF) microscopy combined with direct stochastic reconstruction techniques[23,24], fluorescence-fluctuation-based techniques[25], and wide field-of-view waveguide-based points accumulation in nanoscale topography (PAINT)[26]. The operating wavelength range of current PICs extends from visible to mid-infrared, but does not yet includes the UV range extending from 200 to 400 nm. As discussed below, we have filled this gap by developing UV-PICs operating with low-loss single-mode waveguides at 360 nm. These UV-PICs provide high-visibility, and phase-controlled structured illumination patterns that are located in the far-field above the photonic chips with an excitation numerical aperture ($NA_{ex}$) as high as 0.9 and switchable over three different orientations in a given plane of, for instance, an object to be imaged. Considering that the typical size of the features of the different building blocks constituting a photonic integrated circuit scales with the wavelength, any residual disorder has a stronger impact on the performance of PICs operating in the UV range. Therefore, the demonstration of UV-PICs in demanding applications such as SIM, as performed here, represents a technological leap forward. SIM has been demonstrated in the far-field[2,4–6] and the near field[27] with bulk optics. It has been investigated in the near field of an integrated photonic circuit operating initially in the red part of the visible spectrum ($\lambda = 660$ nm)[13], and more recently with green laser light ($\lambda = 532$ nm)[28]. Here, we show that the far-field SIM technique can also take advantage of a PIC, especially in the UV wavelength range for label-free samples.

## Results

### UV-PICs for structured illumination

Our UV-PICs are made of aluminum oxide $AlO_x$ layers deposited with an atomic layer deposition (ALD) technique on thermal oxide. In view of recent progress in the processing of $AlO_x$, this material is relevant to

be used as the core material for passive single-mode waveguide, especially since it is compatible with 200 or 300 mm-diameter thermal oxide silicon wafers used in CMOS production lines. Propagation losses as low as 4 dB/cm at $\lambda = 250$ nm have been reported in alumina guiding films deposited on a fused silica substrate with ALD[29] and more recently, West et al. have reported losses lower than 3 dB/cm in single-mode $AlO_x$ waveguides with $SiO_2$ top-cladding at $\lambda = 370$ nm[30]. We have now reached propagation losses for transverse electric (TE) polarized light as low as 3 dB/cm with air top-cladding at $\lambda = 360$ nm in 700 nm-wide waveguides fully etched in 120 nm-thick $AlO_x$ layers, see section Methods. With such dimensions, the $AlO_x$ waveguide propagates a single mode as confirmed by simulations based on an eigenmode solver, and by the general operation of the photonic circuit discussed below.

An optical microscopy image of a typical complete circuit is presented in Fig. 1(a). In addition to single-mode waveguides, it includes three pairs of grating out-couplers (GC), three 50:50 multimodal interferometers (MMI) beam splitters, 1 mm-long adiabatic tapers and Ti/Au-based thermal-phase shifters (PS). The adiabatic tapers expand the width of the input single-mode waveguides from 0.7 μm to a width of 20 μm corresponding to that of the grating out-couplers. The two 1000-cycles-long gratings forming each pair are separated by a distance of 2.8 mm that is optimized to generate a structured illumination pattern sufficiently far away from the chip, over a sufficiently large field of view and with an irradiance compatible with the detection limit of our image sensors (340UV-USB, Thorlabs or QImaging Retiga R3). With the current design a uniform illumination over a field of view (FoV) of 32 μm by 32 μm is achieved, see Fig. S1 in supplementary section and discussion in section Methods. All grating out-couplers are shallowly etched with a nominal depth of 30 nm. They have the same grating pitch $\Lambda_G$: either $\Lambda_G = 180$ nm or $\Lambda_G = 150$ nm corresponding to photonic chips with a designed excitation numerical aperture $NA_{ex} = 0.5$ or $NA_{ex} = 0.9$, respectively. For $NA_{ex} = 0.5$ and a laser excitation wavelength $\lambda_{ex} = 360$ nm, the experimental radiant intensity per grating is

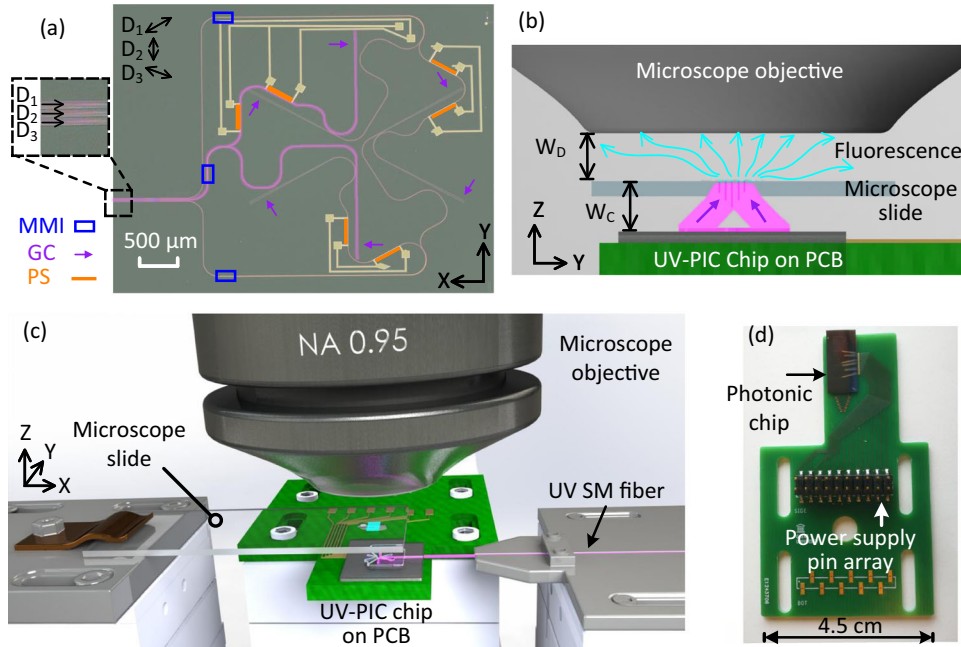

**Fig. 1 | Working principle of the UV-PIC SIM. a** Picture of a UV photonic integrated circuit for structured illumination microscopy. MMI: multimodal interferometer (blue), GC: grating out-coupler (violet) and PS: thermal-phase shifter (orange). $D_1$, $D_2$, and $D_3$ indicate three different optical paths corresponding to structured illumination directions turned at an angle of 120 degrees. Light purple path: $D_2$. **b** Working principle of the UV-PIC-based SIM technique. $W_D$: working

distance of the microscope objective, $W_c$ distance between the top surface of the photonic chip and the sample. The blue and violet arrows illustrate the fluorescence and the UV excitation light, respectively. **c** Schematic of the chip-based far-field SIM set-up including a conventional microscope. **d** Picture of the photonic chip mounted on an electric printed circuit board with gold wire connections.

0.27 mW, which implies that the value of the guided intensity at the entrance of the chip is between 1.8 mW and 7.7 mW based on the theoretical grating coupling efficiency and the fiber-to-chip coupling loss uncertainty, see section Methods. The radiant intensity per grating is currently limited by the maximum power of 24 mW at the input of the optical fiber used to couple the light into the photonic chip, see section Methods.

Each pair of grating out-couplers produces a fringe pattern of coherent light along a given direction $D_i$, with $i = 1$, 2, or 3, in a plane of the far-field located at a distance $W_F$ from the photonic chip. Note that the distance $W_C$ from the surface of the chip to the object is slightly different from $W_F$ due to the refraction of the microscope slide, see Fig. 1(b). The path designed to deliver a structured illumination along direction $D_2$ is highlighted with a thick light purple line. The three fringe patterns are symmetrical by a rotation of 120 degrees around the axis passing through the barycenter of the gratings and perpendicular to the surface of the photonic chip. They are switched on one after the other by mechanically shifting the light coupling fiber at the input of the chip. The phase shifters control the relative phase of the beams scattered by each of the gratings of a given pair and, consequently, enable to spatially shift the pattern of fringes in the far-field along the associated direction $D_i$. For $NA_{ex} = 0.5$, the averaged fringe visibilities in the directions $D_1$, $D_2$, and $D_3$ over the 32 μm by 32 μm field of view are 0.883, 0.889, and 0.911, respectively, with a standard deviation smaller than 0.004, see Fig. S1 in supplementary materials. Ideal visibilities of 1 are not reached due to residual disorder at the gratings and MMI beam splitters. Nevertheless, our photonic chips allow us to achieve very large fringe visibilities over a relevant FoV, which is crucial for implementing SIM where the contrast of high-frequency features in acquired images is proportional to the visibility of the structured illumination.

The properties of flexible beam manipulation and intrinsic compact size make UV photonic chips compatible with any conventional optical microscope, as illustrated in Fig. 1(c). The chip is mounted on a printed circuit board (PCB) and positioned below the object to image. The object is deposited, in the usual way, on a 515 μm-thick borosilicate microscope slide that is transparent to UV light. The PCB enables the electrical current injection of the thermal heaters and offers a convenient handling of the photonic chip, see Fig. 1(d). For excitation numerical apertures $NA_{ex} = 0.5$ and $NA_{ex} = 0.9$, the distances from the surface of the chip to the object are $W_C = 2.75$ mm and $W_C = 1.33$ mm, respectively. The emission from the laser source at the excitation wavelength $\lambda_{ex} = 360$ nm is butt coupled to the chip via a cleaved UV fiber (SM300) mounted on a translation stage. The microscope slide is fixed on a translation stage as commonly used and placed at the imaging plane below the microscope objective with a spacing $W_D$. The part of the set-up above the microscope slide, including the microscope slide itself, is standard for all optical microscopes. The fluorescence of the object that is excited by the UV coherent structured illumination produced by the photonic chip is here collected by a microscope objective of high numerical aperture $NA_{col} = 0.95$ or $NA_{col} = 1.32$. In the following, the first case ($NA_{col} = 0.95$), is used to quantify the optical resolution enhancement of the UV-PIC-assisted SIM. Next, the second case ($NA_{col} = 1.32$) allows us to demonstrate the possibility of using oil-immersion microscope objectives and the viability of our approach for super-resolved imaging with label-free biological samples.

## Resolution enhancement of UV-PIC-based SIM

To evaluate the performance of the UV-PIC, we have investigated the spatial frequency bandwidth of the chip-based SIM. Optical systems that image an incoherent light field are generally characterized by their optical transfer function (OTF), namely the complex transfer function between the normalized spatial frequency spectra of the light intensity fields at the object and image planes[31]. The largest spatial frequency

$K_{Max}$ of the intensity field that can be transmitted through the optical system defines the transmission bandwidth. For a conventional wide-field fluorescent microscope, $K_{Max}^{WF} = \frac{2\pi}{\lambda_{em}} 2NA_{col}$, with $\lambda_{em}$ the wavelength of the field emitted by the object and $NA_{col}$ the numerical aperture of the collecting lens. As a result, the optical spatial resolution is defined as $\frac{\lambda_{em}}{2NA_{col}}$. In the case of structured illumination microscopy, the quadratic nature of the fluorescence process induces a mixing between the spatial frequencies of the exciting coherent field and those of the object. It follows that the largest spatial frequency of the imaged field is $K_{Max}^{SIM} = \frac{2\pi}{\lambda_{em}} 2NA_{col} + \frac{2\pi}{\lambda_{ex}} 2NA_{ex}$ when the fluorescent intensity is linearly proportional to the intensity for excitation field, i.e., for excitation intensities much smaller than the saturation intensity of the fluorescent molecules. The optical spatial resolution can consequently be expressed as $\frac{\lambda_{em}}{2NA_s}$, with the ideal synthetic numerical aperture given by $NA_s = NA_{col} + \frac{\lambda_{em}}{\lambda_{ex}} NA_{ex}$. Note that there is a trade-off between the synthetic aperture $NA_s$ and the field-of-view as $NA_s$ depends on the collection numerical aperture $NA_{col}$ and as higher numerical apertures of the collecting lens are generally accompanied by smaller the field-of-views. From another perspective, the large FoV of collecting lens of moderate $NA_{col}$, such as for instance the typical FoV of 500 by 500 μm for $NA_{col} = 0.5$, can be reached with a synthetic aperture $NA_s = 1$ thanks to the PIC-based SIM technique. Achieving FoVs as large as 16 mm² with a $NA_{col} = 0.25$ as in ref. 7, and here with the corresponding theoretical synthetic aperture $NA_s = 1.52$, will require, as another trade-off, to engineer at least six different values of $NA_{ex}$ to completely fulfill the spatial frequency bandwidth.

To determine the spatial frequency bandwidth, we have processed a sector star target (spoke target) made of a 100 nm-thick gold titanium layer directly deposited on the borosilicate microscope slide previously mentioned, see Fig. 2(a) and scanning electron micrograph in Fig. 2(b). The pattern has been defined by using an electron beam lithography system followed by a lift-off process. This object gives access to a continuous sweep of the spatial frequency of the input light field from its outer part to its inner part, with a line spacing lower than 100 nm at the center. Note that decreasing further the line spacing isotropically is technologically demanding with the current gold thickness. Sector star targets are commonly used with white light illumination. In the context of SIM, fluorescent emission is required, which is here provided by a mixture of dyes (triphenylmethane dyes with pyrene-based dyes) commonly used in green highlighters. When excited at $\lambda_{ex} = 360$ nm, the fluorescent wavelength is $\lambda_{em} = 511$ nm, see Fig. 2(c). The mixture is coated in the liquid phase on the surface of the sector star target to form a 325 nm-thick solid layer after evaporation of the solvent. The UV light emitted from the photonic chip mainly excites the dyes located between the metal lines, leading to a fluorescent object with varying spatial frequencies. As highlighted in Fig. 2(d), (e), and (f), the fringe pattern of the UV field is mapped on the fluorescent intensity field in the three illumination orientations and Moiré patterns are also present as expected. The modulation period of the fringes of the imaged fluorescence is $\Lambda_{ex} = 362$ nm, which corresponds to an experimental excitation numerical aperture $NA_{ex} = 0.497 \pm 0.002$ in line with the designed value.

Using an extended object such as the current fluorescent sector star target is advantageous compared to conventional single point objects in terms of signal to noise ratio (SNR) to infer the optical resolution, as ideal point sources have a limited number of fluorescent molecules and therefore a limited fluorescence rate. In addition, the phase of the OTF, i.e., the phase transfer function (PTF) is directly retrieved from the distortions of the pattern of lines constituting the sector star target. Such distortions have not been detected here.

In Fig. 3, we compare the images of the fluorescent sector star target in the case of the wide-field (WF) and the SIM configurations. Switching from SIM to WF is possible by enabling only one grating out-coupler to irradiate the object. The SIM image is obtained by recording nine frames that correspond to three different relative phases between

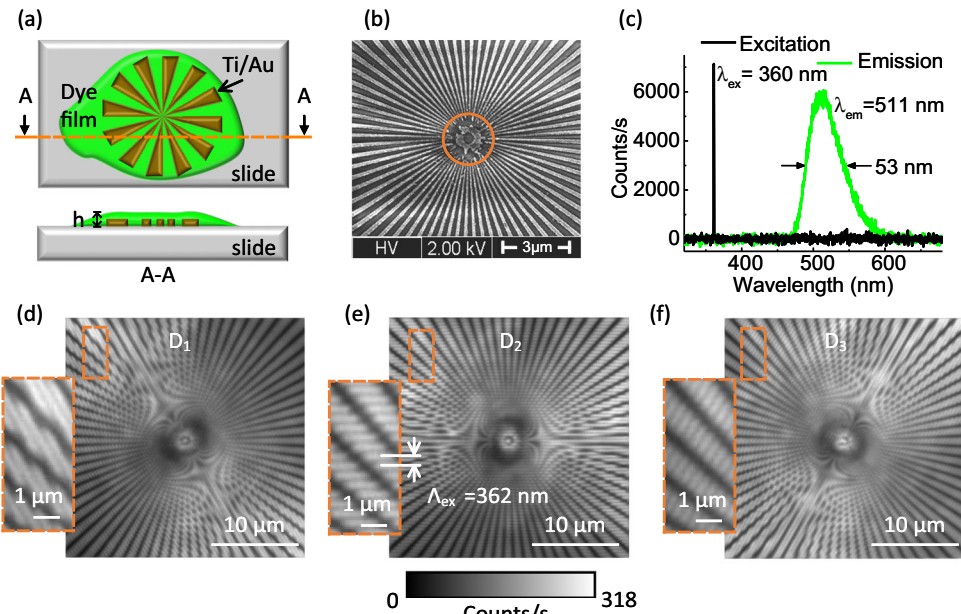

**Fig. 2 | Fluorescent sector star target. a** Schematic of the metal sector star target on borosillicate microscope slide with green fluorescent dyes coated on top. Thickness of the coated dye layer: $h = 300$ nm. **b** Scanning electron micrograph of the metal sector target. The orange solid circle points out the position where the grating pitch along the circle equals to 100 nm. The imaging is independently repeated five times. **c** Optical spectra of the UV exciting beam and the fluorescent dyes. $\lambda_{ex}$: excitation wavelength, $\lambda_{em}$: emission wavelength. **d** to **f** Fluorescence images of the sector target illuminated by structured light for the $D_1$, $D_2$, and $D_3$ orientations, respectively. $\Lambda_{ex}$: modulation period of the fringe pattern.

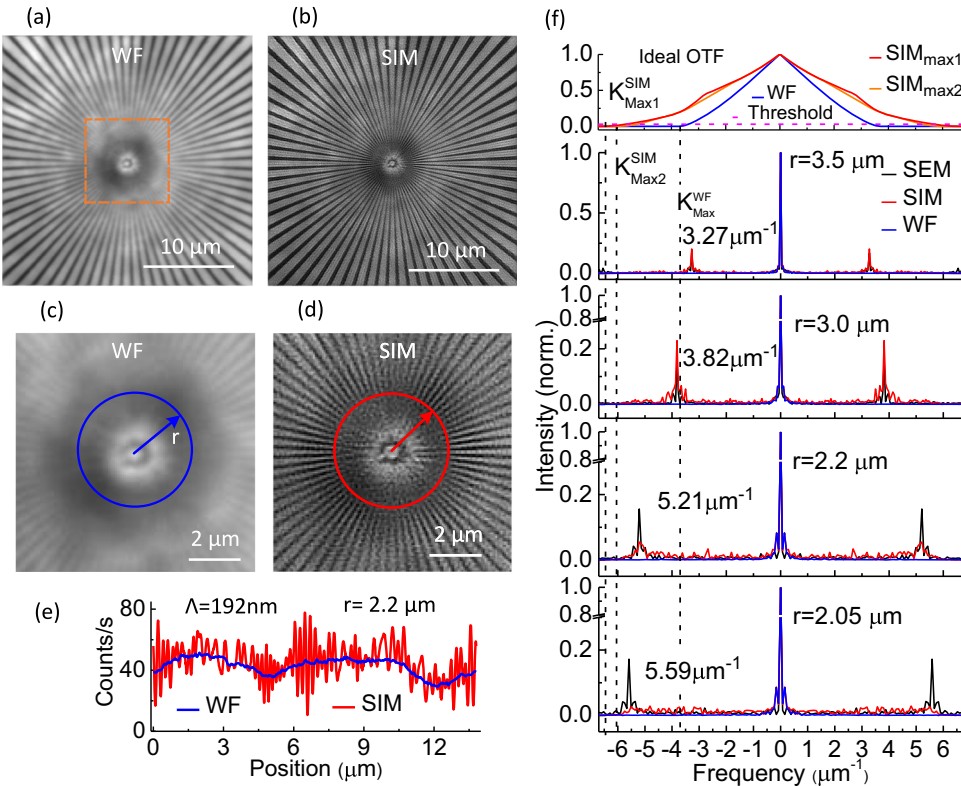

**Fig. 3 | UV-PIC SIM with NA_{ex} = 0.5. a** Standard (raw data) wide-field (WF) image and **b** reconstructed super-resolved SIM image of the fluorescent sector star target in the case of an excitation numerical aperture $NA_{ex} = 0.5$. **c** and **d** Magnified images of the area inside the dashed orange box in **a** and **d**, respectively. The blue and red circles have a radius $r = 2.2$ μm, which corresponds to a grating pitch of 192 nm. **e** Intensity profiles extracted along the circles in **c** and **d**. Λ: spatial period. **f** Fast Fourier transform of the intensity profiles extracted from the WF (blue), SIM (red), and SEM (black) images along circles of radii $r = 3.5$ μm, 3.0 μm, 2.2 μm, and 2.05 μm. The theoretical profiles of the optical transfer function (OTF) for the WF and SIM configurations are plotted at the top where the dashed purple line sets the intensity threshold over which the signal is detectable. The vertical dashed black lines locate the maximum spatial frequencies: $K_{Max}^{WF}$ for the WF configuration, $K_{Max1}^{SIM}$ and $K_{Max2}^{SIM}$ for the SIM configuration, where the two values result from the anisotropy of the OTF.

each pair of grating out-couplers, or equivalently to three different settings of the applied current in the thermal-phase shifters associated with each direction $D_i$. The acquisition time for each frame is 500 ms, which is chosen according to the current SNR of the camera. The reconstruction of the final SIM image makes use of a standard Wiener-filter-based reconstruction algorithm[2,5] with the Wiener parameter set to 0.001. The Wiener parameter $\omega^2$ that is considered as constant and empirically adjusted is defined by the ratio between the intensity of the noise and of the true signal. The reconstruction algorithm requires the knowledge of the OTF of the WF microscope. The OTF of our wide-field microscope has been experimentally estimated by focusing through the microscope objective a laser beam on a mirror and recording the intensity profile at the back-focal plane, which is nothing else than the intensity of the pupil function. Considering that the measured OTF is prone to experimental imperfections coming from the exciting laser beam and is here close to the ideal OTF of a NA = 0.95 collecting lens, we have used the ideal OTF to reconstruct the SIM images, see supplementary information.

Taking the intensity profiles of each image along concentric circles of radius $r$ and of the same center as that of the sector star target, we identify the largest spatial frequency $K_{Max}$ present in the image. The relationship between the $r$ value and the $K$ value is cross-checked with the value of the modulation period of the SEM image (Fig. 2(b)). $K_{Max}$ corresponds to the circular profile for which the frequency peaks of the modulation are drowned in the noise of the fast Fourier transform (FFT) of the profile, see Fig. 3(f). A threshold level of 0.03 in the normalized FFT is used to define the undistinguishable nature of the signal. The red circle in the zoomed SIM image (Fig. 3(d)), for which $r = 2.2\,\mu m$, pinpoints the location of the largest spatial frequency $K_{Max} = 5.2\,\mu m^{-1}$. The associated red profile in Fig. 3(e) exhibits fast variations of spatial period $\Lambda = 192$ nm. In the case of the WF image, the $r = 2.2\,\mu m$ circle is located in a region of the sector star target where the high spatial frequencies of the fluorescent object are not visible. The intensity profile, blue curve in Fig. 3(e), and its FFT in Fig. 3(f), confirm the absence of the fast variations in the detected signal. The experimental optical resolution achieved with the WF configuration is 305 nm. Considering that the optical resolution of our SIM approach is 192 nm, an enhancement factor of 1.58 has been achieved with $NA_{ex} = 0.5$.

The theoretical values of the optical resolution are 269 nm and 154 nm in the WF and SIM cases, respectively, which implies an ideal enhancement factor of 1.75. The difference between the experimental and theoretical optical resolutions is attributed to the current SNR. Using a deep-cooled scientific camera, that is generally used in SIM microscopes, instead of the more conventional and low-cost 20 °C-cooled CCD camera that is used here, will reduce this difference. As a salient feature, our results highlight that even with non-specialized equipment, the UV-PIC can boost the performance of a microscope. Note that by using a 4 mW power laser source reflected on a mirror as mentioned above to retrieve the MTF, the difference between the theoretical and experimental resolution is only 3%.

With Fig. 3(f) we stress the relative position of the ideal optical transfer functions and of the experimental transmitted spatial frequency spectra of intensity profiles at several $r$ values for the WF (blue curves) and SIM (red curves). As revealed in Fig. S2(c), the optical transfer function is not isotropic as only three different orientations are used for the structured illumination. The OTF bandwidth is maximum in the direction of the structured illumination, with the limit $K_{Max1}^{SIM} = 6.49\,\mu m^{-1}$, and minimum in the perpendicular direction, with the limit $K_{Max2}^{SIM} = 5.83\,\mu m^{-1}$, see red and orange curves in Fig. 3(f), respectively. When averaged over all directions, the largest spatial frequency retrieved in the SIM image is close to $K_{Max2}^{SIM}$. The identification of the frequency peaks is confirmed with the associated FFT of the circular profiles of the SEM image, in particular for the largest spatial frequency where the radius is $r = 2.2\,\mu m$.

To illustrate the flexibility of UV-PICs, we have designed chips for which the grating out-couplers enable an excitation numerical aperture as large as $NA_{ex} = 0.9$. The theoretical enhancement factor $1 + \frac{\lambda_{em}}{\lambda_{ex}}\frac{NA_{ex}}{NA_{col}}$ of the optical resolution amounts to 2.35 for $NA_{ex} = 0.9$, $NA_{col} = 0.95$ and the current excitation and emission wavelengths. Using index-matching liquids between the chip and the borosilicate microscope slide, the refractive index of which ($n = 1.54$) is smaller than that of aluminum oxide ($n = 1.69$) at $\lambda_{ex} = 360$ nm, even enhancement factors as large as 3.3 are in principle reachable. Note also that the larger the emission wavelength relative to the excitation wavelength, the larger the enhancement factor.

The fringe patterns obtained with the $NA_{ex} = 0.9$ gratings are free of distortion for the three directions of illumination, as shown in Fig. 4.

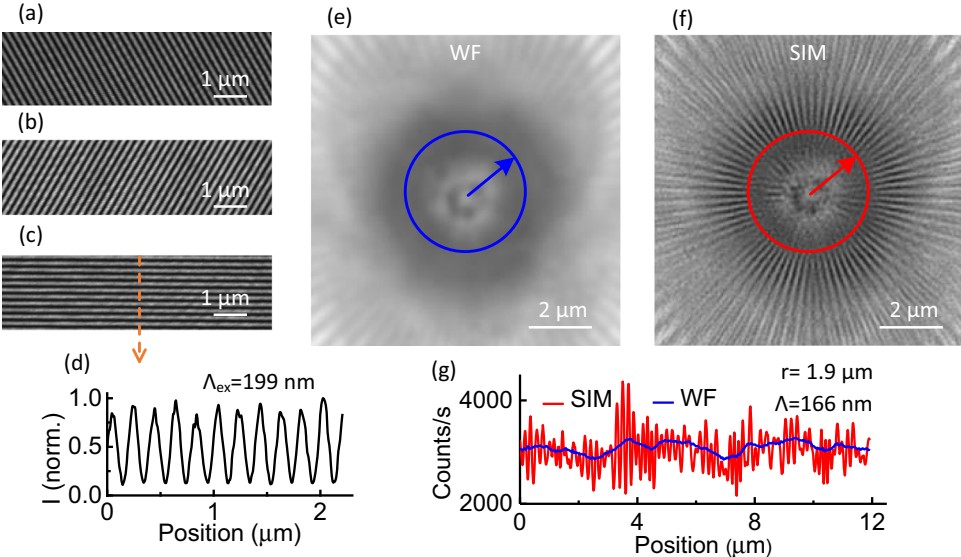

**Fig. 4 | UV-PIC SIM with $NA_{ex} = 0.9$. a–c** Optical images of the UV interference fringe patterns in the case of $NA_{ex} = 0.9$ for the $D_1$, $D_3$, and $D_2$ illumination orientations, respectively. **d** Normalized intensity profile along the dashed line in **c**. $\Lambda_{ex}$: modulation period of the fringe pattern. **e** and **f** Standard wide-field (WF) image and reconstructed SIM image of the fluorescent sector target excited with the patterns in **a–c**. The radii of the circles $r$ are equal to 1.9 μm, which corresponds to a period of the spatial modulation $\Lambda$ of 166 nm. **g** Intensity profiles along the circles in **e** and **f**.

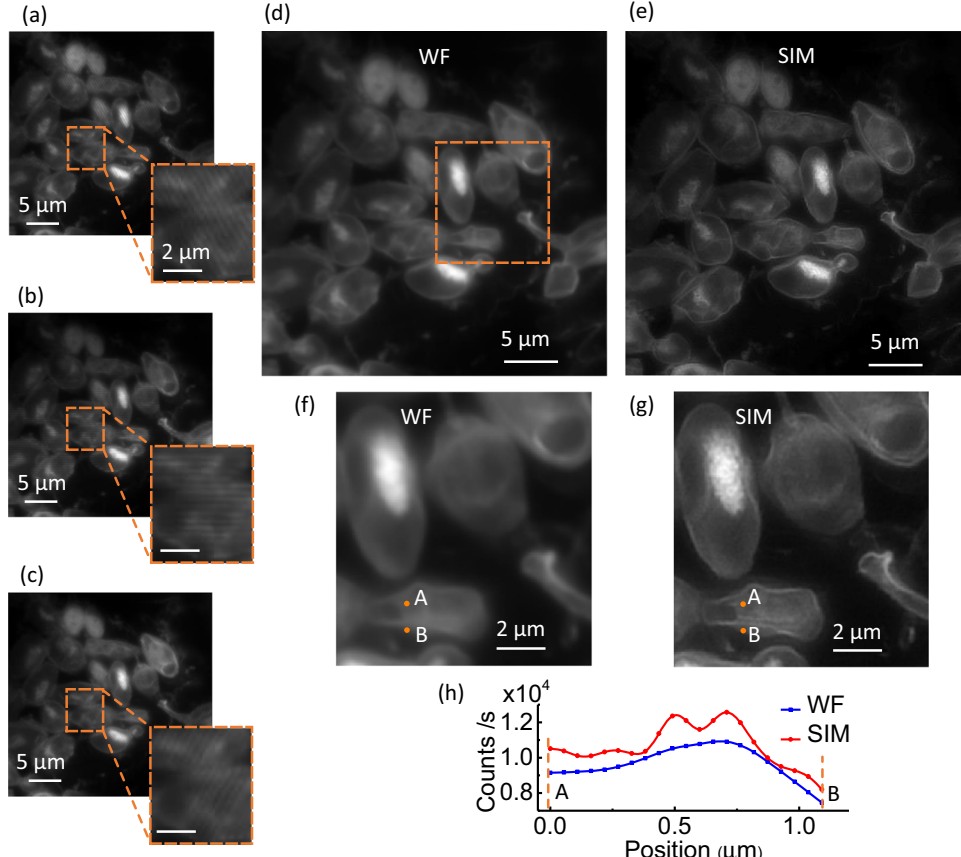

**Fig. 5 | UV-PIC SIM on yeast cells. a–c** Autofluorescence image of NADH in yeast cells under UV structured illumination with $D_1$, $D_2$, and $D_3$ orientations, respectively. The insert is a three times magnified zoom of the area defined by the orange dotted contour to highlight the modulation patterns of the fluorescence intensity resulting from the structured illumination. **d** Standard wide-field and **e** reconstructed SIM images of NADH in yeast cells. The experiment is independently repeated three times for **a**–**e** on different samples. **f** and **g** zoomed images of the area delimited by the orange rectangle in **d** and **e**, respectively. **h** Cross-section profiles along the segment [AB] in **f** and **g**.

The fringe spacing of 199 nm is in line with the designed numerical aperture. The visibilities are slightly lower than that of the case of $NA_{ex} = 0.5$, with values of 0.723, 0.774 and 0.737 for orientation $D_1$, $D_2$, and $D_3$, respectively. The SIM imaging with the $NA_{ex} = 0.9$ illumination is able to resolve a sector spacing of 166 nm over the intensity profile of an entire circular cross-section of radius $r = 1.9\ \mu m$, implying an optical resolution enhancement factor of 1.84. Even if the enhancement factor is larger than for the case of $NA_{ex} = 0.5$, the difference with the expected theoretical value (2.35) is quite significant. In addition to the contribution of the limited SNR of the camera, we attribute this difference to a less dense and less homogeneous distribution of the fluorescent dyes when the spacing between the sectors shrinks. The possibility to optically resolve the sector spacing for some limited parts of the intensity profiles of the circular cross-sections when the radius is lower than $r = 1.9\ \mu m$ supports this last point.

**Imaging biological cells with the UV-PIC-based SIM**
Having assessed the performance of the UV-PIC for structured illumination microscopy with an artificial sample made of exogenous fluorescent dyes, we apply our technique to a label-free biological sample. As, unlike an artificial sample, there is no a priori knowledge about the dimensions of the biological object, the increased quality of the SIM images is revealed by comparing them with those obtained with the standard WF microscopy. In Fig. 5, we compare WF and SIM images of the autofluorescence of yeast cells (IHEM 3961) that are model organisms of interest for biological research. The autofluorescence is mainly provided by the intracellular coenzyme NADH. This molecule exhibits a strong absorption at the excitation

wavelength $\lambda_{ex} = 360$ nm and provides fluorescence in the blue with a peak maximum at $\lambda_{em} = 480$ nm. The UV-PIC designed for $NA_{ex} = 0.5$, which exhibits the higher visibility, is used to provide the UV structured light and the autofluorescence is collected via an oil-immersion microscope objective with $NA_{col} = 1.32$ in order to maximize the collected signal. Note that the endogenous fluorescence is much weaker and more prone to fast quenching than well-engineered exogenous dyes. The use of immersion oil also minimizes phase jumps at the interfaces between the cells and the surrounding medium, which otherwise lead to distortion of the structured illumination and produce artifacts in the reconstructed SIM images. To avoid any damage induced by the UV light[32], the irradiance is set to a value of 3 W/cm². This light intensity is relatively low compared with typical values of $1 - 10^2$ W/cm² for SIM microscopy. Each image is acquired over an integration time of 4 s with a – 20 degrees Celsius cooled CCD camera (Retiga R3, Qimaging) using the software μManager 2.0. In such experimental conditions, the autofluorescence clearly reveals not only the presence of cells but also a mapping of the underlying structured illumination without any deformation, see zoom boxes in Fig. 5(a), (b), and (c). The periodic modulation of the fluorescence with a 362 nm period can indeed be observed here as $NA_{col} > NA_{ex}$. The low-fluorescence quantum yield $\Phi = 0.019$ of the NADH molecules[33] compared to that of fluorescent probes such as green fluorescent molecules for which $\Phi = 0.8$ explains the relatively long exposure time. Like the majority of biological fluorophores, NADH molecules are also prone to photobleaching: the intensity of the NADH autofluorescence is decreased by a factor 2 after 4 min of continuous 30 W/cm² UV excitation. Using a deep-cooled scientific

camera is expected to further decrease the acquisition time down to tens of milliseconds.

Shifting the phase of the structured illumination and implementing image reconstruction, we obtained the super-resolved SIM image in Fig. 5(e), a part of which is zoomed in Fig. 5(g). Compared with the WF image in Fig. 5(d), more biological features are present in the SIM image: dual membranes of cells are clearly visible in the SIM image (see Fig. 5(g)) but indistinguishable in the WF image (see Fig. 5(f)). The theoretical diffraction-limited resolution for the standard wide-field microscopy image is 182 nm whereas it drops to 121 nm in the SIM case, which corresponds to a theoretical 1.5 resolution enhancement in favor of the SIM image. Using the Fourier ring correlation method[34], see supplementary section, the wide-filed and SIM resolutions are 262 nm and 166 nm, respectively. The expected resolution enhancement results in the appearance of two peaks in the profile of the straight A to B cross-section of the SIM image (see Fig. 5(h)), with a peak contrast of 0.03.

## Discussion

This last result demonstrates that the UV-PIC SIM is an useful approach for super-resolution label-free imaging of crucial biological samples. Importantly, the technique does not modify the standard usage protocols for biological wide-field fluorescence imaging as the photonic chip is decoupled from the sample and from the collecting lens of the microscope: the UV-PIC is a plug and play add-on. Whether the setup can be built cost-effectively is one of the most frequently asked questions for the SIM technique[35]. Moreover, it is a burning issue for any kind of UV-based microscopy techniques, as UV-enhanced components are much more expensive than those at visible wavelengths. The proposed UV-PICs are expected to provide a solution to this issue, in view of their robustness, multi-functionalities and compatibility with large-scale fabrication methods.

Similarly to the wide-field UV hyper-spectral imaging reported in[1], using a structured illumination at $\lambda = 360$ nm also opens the door for combining the label-free and super-resolution modalities with a hyperspectral modality considering that a large amount of biological molecules, such as the collagen ($\lambda_{em} = 393$ nm), elastin ($\lambda_{em} = 410$ nm) and flavin adenine dinucleotide ($\lambda_{em} = 520$ nm), exhibit autofluorescence at this excitation wavelength. Extending further the hyper-spectral imaging with a broad multi-color illumination is challenging with a single UV circuit. The proposed waveguide geometry is guiding a single mode from a wavelength of 320 nm up to a cut-off at 420 nm, which hinders the simultaneous use of visible wavelengths, for instance at 532 nm or 632 nm. The implementation of broad multicolor illumination is all the more challenging with the same circuit that the beam splitters and grating out-couplers are wavelength dependent and here optimized for a wavelength of 360 nm. Nevertheless, a possible approach to enable a broad multi-color excitation consists in patterning different circuits in multi-layer stack of planar waveguide made of $AlO_x$ for UV and silicon nitride (SiN) for visible light, in the same way as in[36]. Besides, in view of the high transmittance of the aluminum oxide material down to a wavelength of 250 nm, the proposed UV-PIC SIM technique is also a promising avenue for deep-UV super-resolved fluorescence microscopy. Finally, following the progress already made at near-infrared and visible wavelengths, one of the remaining great challenges for the future is to directly integrate coherent UV sources with the photonic chip in a compact way.

To conclude, we have used UV $AlO_x$-based photonic integrated circuits as an add-on module to boost the performance of conventional fluorescence microscopes. With well-controlled chip-based structured illuminations and a well-defined fluorescent sector star target, we have characterized the resolution limit of the UV-PIC SIM. Starting from a microscope objective of NA = 0.95, the UV-PIC has allowed us to synthesize a numerical aperture value up to NA = 1.75, resulting in an optical resolution enhancement by a factor of 1.84

compared to a conventional fluorescence microscope. As a highlight, our UV-PIC chip has enabled to observe, in a label-free, far-field and wide-field configuration, features in yeast cells that are otherwise indistinguishable with standard wide-field microscopy. This UV-PIC has a strong potential for improving pathology inspections and diagnostics, for triggering the discovery of therapeutic targets, and consequently for enhancing throughput of drug development.

## Methods

### Chip fabrication

The UV-PICs are made as follows. Step 1: a layer of 120 nm $AlO_x$ is grown by a thermal ALD technique on top of a 3 μm-thick thermal $SiO_2$ oxide on a silicon wafer. The ALD deposition is carried out at a deposition temperature of 300 °C, at a base pressure of $10^{-6}$ mbar and precursor pressures of $5 \times 10^{-3}$ mbar, using trimethylaluminum (TMA) and $H_2O$ as precursors, with a growth rate of 1 Å per cycle. Step 2: a silicon nitride ($SiN_x$) hard mask is deposited by plasma enhanced chemical vapor deposition (PECVD). Step 3: the pattern defined by E-beam lithography in a coated resist film is transferred to the hard mask through reactive ion etching (RIE). Step 4: the alumina layer is fully etched by using inductively coupled plasma-RIE in a gas mixture of $BCl_3/Cl_2/Ar$ to define the UV-PICs. Repeating steps 2 to 4, the grating out-couplers of 30 nm depth are shallowly etched on expanded 20 μm-wide alumina waveguides. Step 5: the $SiN_x$ hard mask is removed by RIE due to its strong absorption at UV wavelengths. Step 6: thermal-phase shifters are fabricated by processes of E-beam lithography, metallization and lift-off. Step 7: the photonics chip is fixed on a PCB and connected to contact pads via wire-bonding. The waveguide dimensions are designed for a width of 700 nm and a height of 120 nm compatible with a single-mode propagation at $\lambda_{ex} = 360$ nm. The 142 μm-long MMI is optimized to split equally the guide mode into two paths. A 1 mm-long adiabatic taper expanding the waveguide from a width of 700 nm to 20 μm has been optimized via finite-difference time-domain (FDTD) simulations in Lumerical. A designed duty cycle of 0.8 has been chosen to minimize the scattering at the grating input.

### Sample preparation

The yeast strain is ordered from DSMZ (Braunschweig, German) and cultured for 24 h in nutrient broth at 28 °C. A droplet of the yeast solution is taken and dropped on the glass substrate. The sample is ready for imaging after oil immersion and without using any coverslip. The low-autofluorescence immersion oil (IMMOIL-F30CC, Olympus) is chosen to minimize the autofluorescence background at UV excitation.

The metal sector star target is processed with E-beam lithography, E-beam physical vapor deposition, and lift-off. To label the metal sector star target, a mixture solution of dyes is extracted from green highlighters (STABILO, German). After dilution in deionized water in a ratio of 1:3, the dye solution is dropped on the metal sector star target and a thin layer forms after the water evaporates.

### Measurement

A continuous-wave UV solid state laser source (CNIlaser, UV-F-360) with a 360 nm emission wavelength generates the UV light into the PICs. The maximum laser output power is 50 mW. The power drops to 24 mW after coupling to the single-mode optical fiber (SM300) used to excite the photonic chip. The maximum output power of 0.27 mW per grating out-coupler implies that the total insertion loss of the PIC is equal to 16.5 dB. This total loss comes from the fiber-to-chip coupling loss of theoretical value ~5 dB, the grating out-coupler efficiency of theoretical value ~2.5 dB, the 1 cm-long routing single-mode waveguide ~3 dB and residual disorder at the MMI, the waveguide splitter and the gratings. Assuming the theoretical values for the fiber-to-chip coupling loss and the grating out-coupler efficiency, the loss value due to the residual disorder is ~6 dB, leading to a guided intensity as high

as 7.7 mW at the beginning of the single-mode waveguide. This value drops to 1.8 mW if the impact of the residual disorder is negligible. As a result, a guided intensity of 1.8 mW does certainly not damage the PICs at a wavelength of 360 nm and the same is possibly true for a guided intensity of 7.7 mW. Finally, to filter the UV laser and collect only the fluorescence signal, a long-pass filter (Semrock, FF01-380/LP) is inserted into the imaging path. The magnification of the entire imaging system is 83 × and 66 × for the objective lens of numerical aperture 1.32 and 0.95, respectively. With the physical pixel size of 4.54 μm at the camera imager, the sizes of the optical sampling in the plane of the object are 54.6 nm and 68.6 nm, respectively. The optical resolutions are determined to be 269 nm and 182 nm for $NA_{col} = 0.95$ and $NA_{col} = 1.32$, respectively, i.e., more than two times the optical sampling size, which assures that the Nyquist criterion is fulfilled.

The field-of-view is here limited by the dimension of the radiated field in the direction of the length of the grating. It is possible to achieve a larger FoV by increasing the effective propagation length of the mode guided inside the grating, namely by decreasing the etch depths of the grating spatial modulation and by further increasing the grating length. We have fabricated samples with a shallow grating etch depth of 10 nm that resulted in field of views of 150 by 150 μm while maintaining a high excitation numerical aperture $NA_{ex}$. However, the larger the FoV, the lower the irradiance for the same UV intensity coupled into the chip. The current FoV of 32 by 32 μm has been chosen to optimize the signal to noise ratio of the images with our available UV power budget. The ultimate upper limit of the field of view in our far-field set-up is set by the collecting lens, which amounts for instance to a diameter of 160 μm for our $NA_{col} = 0.95$ microscope objective.

For shifting the phase, three different voltages, namely 0 V, 9 V, and 15 V are set in view of targeting phase shifts of 0, $\frac{2\pi}{3}$ and $\frac{4\pi}{3}$, respectively. Owing to the low thermo-optic coefficients of $AlO_x$, the required applied voltages are relatively high compared to those used in silicon phase shifters, and the associated local thermal perturbations lead to slow thermal drifts for non-thermally stabilized photonic chips. In general on-chip thermal-phase shifters can response in sub-milliseconds for each phase stepping, when temperature controllers are used on the backside of the photonic chip[37]. The current chip is not thermally stabilized as usually done with a Peltier element in order to minimize the overall electrical power consumption, which results in a low accuracy of the phase value adjustment as a trade-off. The actual experimental value of the phase is accurately retrieved by using a correlation-based iterative algorithm for reconstruction (see the code in Supplementary Dataset 1)[38]. As an example, the three phases retrieved for the first orientation $D_1$ in the case of the sector star target are 0, 89, and 192 degrees with a precision of ±2 degrees with the current signal to noise ratio.

## Reporting summary

Further information on research design is available in the Nature Research Reporting Summary linked to this article.

## Data availability

The source data for Figs. 3 and S2 are available in the Supplementary Dataset 1 and other data that support the findings of this study are available at figshare https://doi.org/10.6084/m9.figshare.20015132.

## Code availability

The reconstruction codes are available in the Supplementary Dataset 1. The code has been tested on the software Matlab 2018a.

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

## Acknowledgements

N.L.T. acknowledges the INTEC Department of Ghent University for financing the purchase of the UV camera. The research is supported by Methusalem Grant (Flemish Government) R.B., Bijzonder Onderzoeksfonds Interdisciplinair Onderzoeksproject (BOF-IOF) (01IO1320) N.L.T., and FWO-onderzoeksproject WEAVE (G033722N) N.L.T.

## Author contributions

C.L. and N.L.T. conceived the experimental set-up. C.L. designed and fabricated the samples, and performed the experiment. C.L. and N.L.T. analyzed the data. J.P. and D.S. optimized the material deposition. N.B. provided biological samples. C.L. and N.L.T. wrote the main part of the manuscript. J.D. and C.D. supervised the ALD deposition. J.D., C.D., and R.B. commented on the manuscript. N.L.T. supervised the research project.

## Competing interests

The authors declare no competing interests.
