## [Peer Review File · Nature Communications]

UV photonic integrated circuits for far-field structured illumination autofluorescence microscopyREVIEWER COMMENTS

Reviewer #1 (Remarks to the Author):

In this article Chupao Lin et. al. proposed the development of alumina oxide waveguide platform to deliver photonic-chip assisted structure illumination microscopy at UV wavelength. Aluminum oxide waveguide material is interesting as it supports broad range of spectrum down to UV. Photonic integrated circuits (PICs) have been recently being reported to perform light beam shaping for super-resolution optical microscopy methods. This research direction is relatively new and is very attractive as it holds the potential to miniaturizes the bulky microscopy set-up and integrate different imaging and sensing modalities. In this article authors demonstrated free-space light beam shaping by generating interference fringes at UV wavelengths using an array of optical waveguide. Further by using thermo-optical phase modulation and 3 orientation of waveguide array successfully demonstrated 2D SIM at UV wavelengths by out-coupling the guided light inside the waveguide. The proof-of-principle is demonstrated by performing label-free super-resolution imaging of yeast cells. This research work is interesting and pioneer in demonstrating label-free UV-SIM using PICs and thus I recommend paper for Nature communication after following revisions are made:

A) Could authors elaborate the discussion on how to increase to enhance the FoV to more reasonable numbers e.g. 500x500 μm^2 or larger. Will it be possible to support large N.A. illumination for such a large FoV? What are the trade-offs of FoV and effective N.A. for illumination.

B) In microscopy, multi-color imaging is routinely used. Also in the conclusion section authors discuss about hyper-spectral imaging. How will the proposed grating geometry work for different wavelength? Author should provide discussions related to the challenges and opportunities to use grating out-coupling for broad multi-color imaging (e.g. 365-660 nm, regime). Perhaps a simulation studies in the supplementary on possible designs and geometries that could support multiple wavelength illuminations using the proposed concept will be useful for the readers.

C) The acquisition time was very high for biological imaging, 5s for biological samples and 0.5s for fluorophores. Typically, 10-100ms exposure is sufficient, was this due to low auto-fluorescence from the sample or low intensity supported by the chip. It will be nice to discuss this further.

D) A discussion about thermo-phase shifting is required in the paper, e.g. how fast this was accomplished and discussion about precision and accuracy of the phase stepping would be useful for the readers. Did authors collected 3 specific phase steps or a series of phase steps and then extracted 3 uniform spaced phase steps?

E) Details about magnification of the objective lens were missing, only N.A. was mentioned. What was the total magnification of the system and the final pixel size? Did it fulfill the Nyquist rate for SIM imaging?

F) Authors mentioned that UV-SIM implementation with other configuration will be challenging due to the needs of SLM and DMD operating at UV wavelengths, but there are several other approaches on SIM, that also decouples the illumination and collection light paths and use mirror assembly for delivering SIM (Joby Joseph et al 2020 J. Phys. D: Appl. Phys. 53 044006), these approaches albeit not demonstrated UV wavelengths and does not use SLMs & DMDs. It is recommended that authors compare their work with such previous approaches, in terms of achievable resolution and FoV.

G) Similarly, it is recommended to review different super-resolution optical microscopy using photonic-chip configurations to provide a holistic overview of the field.

H) It is recommended that authors provide raw data and the reconstruction code. Alternatively, authors can demonstrate the artefacts reconstruction using openly accessible SIM reconstruction

software such as FAIR-SIM.

I) Would it be beneficial to add a band-pass filter in the experiments with cells? Perhaps the autofluorescence spectra from yeast cells are broad?

J) In the method section, authors should include the details about sample preparation steps. Was a coverslip used to close the sample stage, which cell medium was used and other details for sample preparation.

Reviewer #2 (Remarks to the Author):

In the manuscript „UV photonic integrated circuits for far-field structured illumination autofluorescence microscopy“ Liu et al. present a novel UV-compatible photonic integrated circuit (UV-PIC). The photonic chip enables structured illumination microscopy for super-resolved autofluorescence imaging of yeast cells. UV-compatible photonic chips overcome the limitations of specialized UV optics since excitation of the sample is performed “on chip”, while standard VIS-range optics can be used for, but not limited to, NADH fluorescence detection. Hence, novel UV-compatible PIC may provide a cost-effective alternative to standard UV fluorescence microscopy that additionally enables super-resolution capabilities.

The paper addresses a broad readership in the field of photonic chips, super-resolution microscopy and UV microscopy. It is very well and clearly written. The technique the authors present is novel and a valuable continuation of the current research on microscopy on photonic chips. The data and analyses presented support the conclusions drawn in the manuscript. The data analysis lacks an established method for determining the optical resolution, such as Fourier ring correlation (see below). The methodology is sound and the design and manufacturing of the PIC is described in details. Additional information on sample preparation is required.

Therefore, the following issues require revision:

Please provide detailed information on labelling the sector star target (Fig. 2) and yeast cells (Fig. 5) preparation in the Methods section. Please also comment on photobleaching effects.

Why is Wiener-filtering applied to the wide-field as well as super-resolved images? In general, Wiener-filtering removes noise but also lowers the optical resolution. Since in the manuscript the resolution of wide-field images is determined or a comparison of wide-field to super-resolution images is performed I would strongly prefer unprocessed (“raw”) data images and analyses performed on these data sets.

In order to determine the optical resolution, the authors should use a standardized method, such as Fourier ring correlation.

Response to reviewers' comments

We would like to thank both reviewers for their pertinent comments. We have taken into account all their remarks by extending the scientific discussion and implementing all their suggestions for modifications. The reviewer's feedback has enriched our manuscript and enabled to clarify some technical aspects. Below, we provide point-by-point response. The reviewers' comments are in black. Responses are in blue. Modifications made in the manuscript are in blue with underscore.

Reviewer #1:

- A) Could authors elaborate the discussion on how to increase to enhance the FoV to more reasonable numbers e.g. 500x500 μm^2 or larger. Will it be possible to support large N.A. illumination for such a large FoV? What are the trade-offs of FoV and effective N.A. for illumination?

Response: The reviewer raises an interesting technical point. We have elaborated the discussion on the field-of-view (FoV) and the trade-offs of effective NA and FoV in the manuscript.

See the added discussion in section methods at Line 377 "The field-of-view is here limited by the dimension of the radiated field in the direction of the length of the grating. It is possible to achieve a larger FoV by increasing the effective propagation length of the mode guided inside the grating, namely by decreasing the etch depths of the grating spatial modulation and by further increasing the grating length. We have fabricated samples with shallow grating etch depth of 10 nm that resulted in field of views of 150 by 150 μm while maintaining a high excitation numerical aperture NA_{ex} . However, the larger the FoV, the lower the irradiance for the same UV intensity coupled into the chip. The current FoV of 32 by 32 μm has been chosen to optimize the signal to noise ratio of the images with our available UV power budget. The ultimate upper limit of the field of view in our far-field set-up is set by the collecting lens, which amounts for instance to a diameter of 160 μm for our $\text{NA}_{\text{col}}=0.95$ microscope objective."

See the added discussion at Line 159 "Note that there is a trade-off between the synthetic aperture NA_s and the field-of-view as NA_s depends on the collection numerical aperture NA_{col} and as higher numerical apertures of the collecting lens are generally accompanied by smaller the field-of-views. From another perspective, the large FoV of collecting lens of moderate NA_{col} , such as for instance the typical FoV of 500 by 500 μm for $\text{NA}_{\text{col}}=0.5$, can be reached with a synthetic aperture $\text{NA}_s=1$ thanks to the PIC-based SIM technique. Achieving FoVs as large as 16 mm^2 with a $\text{NA}_{\text{col}}=0.25$ as in [Ref. Joseph], and here with the corresponding theoretical synthetic aperture $\text{NA}_s=1.52$, will require, as another trade-off, to engineer at least 6 different values of NA_{ex} to completely fulfill the spatial frequency bandwidth."

J. Joseph, K. P. Faiz, M. Lahrberg, J.-C. Tinguely, and B. S. Ahluwalia, "Improving the space-bandwidth product of structured illumination microscopy using a transillumination configuration," *Journal of Physics D: Applied Physics*, vol. 53, no. 4, (2019).

- B) In microscopy, multi-color imaging is routinely used. Also in the conclusion section authors discuss about hyper-spectral imaging. How will the proposed grating geometry work for different wavelength? Author should provide discussions related to the challenges and opportunities to use grating out-coupling for broad multi-color imaging (e.g. 365-660 nm, regime). Perhaps a simulation studies in the supplementary on possible designs and geometries that could support multiple wavelength illuminations using the proposed concept will be useful for the readers.

Response:

We thank the reviewer for this interesting comment. When we referred to the hyper-spectral imaging in the conclusion, we focused only on a spectrally filtered imaging in the collecting path of the auto-fluorescence. We have now added in the conclusion a discussion on the possibility of a broad multi-color imaging.

See the added discussion at Line 307 "Extending further the hyper-spectral imaging with a broad multi-color illumination is challenging with a single UV circuit. The proposed waveguide geometry is for instance guiding a single mode from a wavelength of 320 nm up to a cut-off at 420 nm, which hinders the simultaneous use of visible wavelength, for instance at 532 nm or 632 nm. The implementation of broad multi-color illumination is all the more challenging with the same circuit that the beam splitters and grating out-couplers are wavelength dependent and here optimized for a wavelength of 360 nm. Nevertheless, a possible approach to enable a broad multi-color excitation consists in patterning different circuits in multi-layer stack of planar waveguide made of AlO_x for UV and silicon nitride (SiN) for visible light, in the same way as in [34]. Besides, ..."

See the added comment at Line 303 in the conclusion "Similarly to the wide field UV hyper-spectral imaging reported in [1]"

[1] A. Ojaghi, G. Carrazana, C. Caruso, A. Abbas, D. R. Myers, W. A. Lam, and F. E. Robles, "Label-free hematology analysis using deep-ultraviolet microscopy," *Proc Natl Acad Sci U S A* **117**, 14779-14789, Jun 30, (2020).

[34] W. D. Sacher, Z. Yong, J. C. Mikkelsen, A. Bois, Y. Yang, J. C. C. Mak, P. Dumais, D. Goodwill, C. Ma, J. Jeong, E. Bernier, and J. K. S. Poon, "Multilayer Silicon Nitride-on-Silicon Integrated Photonic Platform for 3D Photonic Circuits," *OSA Technical Digest (online)*, paper JTh4C.3 (2016).

- C) The acquisition time was very high for biological imaging, 5s for biological samples and 0.5s for fluorophores. Typically, 10-100ms exposure is sufficient, was this due to low autofluorescence from the sample or low intensity supported by the chip. It will be nice to discuss this further.

Response: We have discussed further the exposure time in the main text.

Line 277 “The low fluorescence quantum yield $\Phi=0.019$ of the NADH molecules [33] compared to that of fluorescent probes such as green fluorescent molecules for which $\Phi=0.8$ explains the relatively long exposure time. Like the majority of biological fluorophores, NADH molecules are also prone to photobleaching: the intensity of the NADH autofluorescence is decreased by a factor 2 after 4 minutes of continuous 30 W/cm^2 UV excitation. Using a deep-cooled scientific camera is expected to further decrease the acquisition time down to tens of milliseconds.”

[33] C. R. Cantor, P. R. Schimmel, Biophysical chemistry Part 2: Techniques for the study of biological structure and function: W. H. Freeman (First edition), 1980.

Note that the fluorophores that we used, namely triphenylmethane dyes extracted from a green highlighter are known to have extremely low fluorescence quantum yield due to easy vibrational de-excitation [Ref: Barbendure], which explains the 0.5s acquisition time.

J. R. Babendure, S.R. Adams, R. Y. Tsien “Aptamers Switch on Fluorescence of Triphenylmethane Dyes”, J. Am. Chem. Soc. 125, 14716-14717, (2003).

- D) A discussion about thermo-phase shifting is required in the paper, e.g. how fast this was accomplished and discussion about precision and accuracy of the phase stepping would be useful for the readers. Did authors collected 3 specific phase steps or a series of phase steps and then extracted 3 uniform spaced phase steps?

Response: We have added a discussion about the properties of the thermal-phase shifter in the section “methods”.

Line 387. “For shifting the phase, three different voltages, namely 0V, 9V and 15V are set in view of targeting phase shifts of $0, \frac{2}{3}\pi$ and $\frac{4}{3}\pi$, respectively. Due to the low thermo-optic coefficients of AlO_x , the required applied voltages are relatively high compared to those used in silicon phase shifters, and the associated local thermal perturbations lead to slow thermal drifts for non-thermally stabilized photonic chips. In general on-chip thermal-phase shifters can response in sub-milliseconds for each phase stepping, when temperature controllers are used on the backside of the photonic chip [36]. The current chip is not thermally stabilized as usually done with a Peltier element in order to minimize the overall electrical power consumption, resulting in a low accuracy of the phase value adjustment as a trade-off. The actual experimental value of the phase is accurately retrieved by using a correlation based iterative algorithm for reconstruction [37]. As an example, the three phases retrieved for the

first orientation D_1 in the case of the sector star target are 0, 89 and 192 degrees with a precision of ± 2 degrees with the current signal to noise ratio.”

- [36] Z. Yong, H. Chen, X. Luo, A. Govdeli, H. Chua, S. S. Azadeh, A. Stalmashonak, G. Q. Lo, J. K. S. Poon, and W. D. Sacher, “Power-efficient silicon nitride thermo-optic phase shifters for visible light,” *Opt Express* **30**, 7225-7237, (2022).
- [37] A. Lal, C. Shan, and P. Xi, “Structured Illumination Microscopy Image Reconstruction Algorithm,” *IEEE Journal of Selected Topics in Quantum Electronics* **22**, 50-63 (2016).

- E) Details about magnification of the objective lens were missing, only N.A. was mentioned. What was the total magnification of the system and the final pixel size? Did it fulfill the Nyquist rate for SIM imaging?

Response: The information of the magnification of the objective and the pixel size are now added to the section “methods”.

Line 371 “The magnification of the entire imaging system is 83x and 66x for the objective lens of numerical aperture 1.32 and 0.95, respectively. With the physical pixel size of 4.54 μm at the camera imager, the sizes of the optical sampling in the plane of the object are 54.6 nm and 68.6 nm, respectively. The optical resolutions are determined to be 269 nm and 182 nm for $\text{NA}_{\text{col}}=0.95$ and $\text{NA}_{\text{col}}=1.32$, respectively, namely more than two times the optical sampling size, which assures that the Nyquist criterion is fulfilled.”

- F) Authors mentioned that UV-SIM implementation with other configuration will be challenging due to the needs of SLM and DMD operating at UV wavelengths, but there are several other approaches on SIM, that also decouples the illumination and collection light paths and use mirror assembly for delivering SIM (Joby Joseph et al 2020 *J. Phys. D: Appl. Phys.* 53 044006), these approaches albeit not demonstrated UV wavelengths and does not use SLMs & DMDs. It is recommended that authors compare their work with such previous approaches, in terms of achievable resolution and FoV.

Response: We thank reviewer for recommending this interesting paper. Although a SLM is still used in their configuration to control the phase and the orientation of the structured illumination, the concept of using mirrors to decouple the excitation beam from the collection beam is interesting. We have included it in the reference.

Line 48 “Moreover, shaping and steering UV beams is crucial, in particular for structured illumination microscopy (SIM), which was originally implemented with bulky gratings [2,3] and more recently with spatial light modulators (SLM) or digital micro-mirror devices (DMD), but only at wavelengths larger than 365 nm [4-6]. Combining mirrors and a SLM to control”

the phase and the orientation of the structured illumination has afterwards enabled to decouple the excitation and collection in order to enlarge the field of view up to 16 mm² with a collection numerical aperture of 0.25 [7], but still in a bulky configuration and at visible wavelengths.”

- [2] M. G. Gustafsson, “Surpassing the lateral resolution limit by a factor of two using structured illumination microscopy,” *J Microsc*, vol. 198, no. Pt 2, pp. 82-7, May, 2000.
- [3] M. G. Gustafsson, L. Shao, P. M. Carlton, C. J. Wang, I. N. Golubovskaya, W. Z. Cande, D. A. Agard, and J. W. Sedat, “Three-dimensional resolution doubling in wide-field fluorescence microscopy by structured illumination,” *Biophys J*, vol. 94, no. 12, pp. 4957-70, Jun, (2008).
- [7] J. Joseph, K. P. Faiz, M. Lahrberg, J.-C. Tinguely, and B. S. Ahluwalia, “Improving the space-bandwidth product of structured illumination microscopy using a transillumination configuration,” *Journal of Physics D: Applied Physics*, vol. 53, no. 4, (2019).

As already mentioned in point A, we have added in line 163.

“Achieving FoVs as large as 16 mm² with a NA_{col}=0.25 as in [Ref. Joseph], and here with the corresponding theoretical synthetic aperture NA_s=1.52, will require, as another trade-off, to engineer at least 6 different values of NA_{ex} to completely fulfill the spatial frequency bandwidth.”

- G) Similarly, it is recommended to review different super-resolution optical microscopy using photonic-chip configurations to provide a holistic overview of the field.

Response: we have elaborated the discussion about technique of the chip-based super resolution microscopy.

Line 62 “and artificial intelligence [22]. With regard to super-resolution optical microscopy, PICs have been used mainly for near-field imaging, in particular for implementing wide field-of-view waveguide-based total internal reflection fluorescence (TIRF) microscopy combined with direct stochastic reconstruction techniques [23-24], fluorescence-fluctuation-based techniques [25], and wide field-of-view waveguide-based ‘points accumulation in nanoscale topography’ (PAINT) [26].

- [23] R. Diekmann, Ø. I. Helle, C. I. Øie, P. McCourt, T. R. Huser, M. Schüttpelz, and B. S. Ahluwalia, “Chip-based wide field-of-view nanoscopy,” *Nature Photonics*, vol. 11, no. 5, pp. 322-328, 2017.

- [24] J. C. Tinguely, A. M. Steyer, C. I. Oie, O. I. Helle, F. T. Dullo, R. Olsen, P. McCourt, Y. Schwab, and B. S. Ahluwalia, "Photonic-chip assisted correlative light and electron microscopy," *Commun Biol*, vol. 3, no. 1, pp. 739, Dec 7, 2020.
- [25] I. S. Opstad, D. H. Hansen, S. Acuna, F. Strohl, A. Priyadarshi, J. C. Tinguely, F. T. Dullo, R. A. Dalmo, T. Seternes, B. S. Ahluwalia, and K. Agarwal, "Fluorescence fluctuation-based super-resolution microscopy using multimodal waveguided illumination," *Opt Express*, vol. 29, no. 15, pp. 23368-23380, Jul 19, 2021.
- [26] A. Archetti, E. Glushkov, C. Sieben, A. Stroganov, A. Radenovic, and S. Manley, "Waveguide-PAINT offers an open platform for large field-of-view super-resolution imaging," *Nat Commun*, vol. 10, no. 1, pp. 1267, Mar 20, 2019.

Line 78 "It has been investigated in the near field of an integrated photonic circuit operating initially in the red part of the visible spectrum ($\lambda = 660$ nm) [12] and more recently with green laser light ($\lambda = 532$ nm) [28]."

- [12] Ø. I. Helle, F. T. Dullo, M. Lahrberg, J.-C. Tinguely, O. G. Hellesø, and B. S. Ahluwalia, "Structured illumination microscopy using a photonic chip," *Nature Photonics*, vol. 14, no. 7, pp. 431-438, 2020.
- [28] Q. Deng, O. Arisev, D. Kouznetsov, M. u. Hasan, R. Vos, P. Van Dorpe, and N. Verellen, "Structured Illumination Microscopy Based on Silicon Nitride Photonic Integrated Circuits," *OSA Technical Digest*. p. T4D.2 (2022).

- H) It is recommended that authors provide raw data and the reconstruction code. Alternatively, authors can demonstrate the artefacts reconstruction using openly accessible SIM reconstruction software such as FAIR-SIM.

Response: We have submitted the raw data and the reconstruction code in a Zip file.

- I) Would it be beneficial to add a band-pass filter in the experiments with cells? Perhaps the autofluorescence spectra from yeast cells are broad?

Response: We thanks reviewer's insightful advice. We measured the fluorescence spectrum of the autofluorescence of yeast cells and we have now attached it in the supplementary document. The auto-fluorescence spectrum of yeast cells is broad and comes mainly from the NADH with a peak wavelength of 480 nm. Limited by the current experimental signal to noise ratio, the hyper-spectral imaging is still challenging to implement but we plan to study it in future work by using a deep-cooled camera.

Fig. S5 Fluorescence spectrum of yeast cells under UV excitation at \$\lambda = 360\$ nm.

- J) In the method section, authors should include the details about sample preparation steps. Was a coverslip used to close the sample stage, which cell medium was used and other details for sample preparation.

Response: we have included the sample preparation in the method section.

Line 349 “Sample preparation: The yeast strain is ordered from DSMZ (Braunschweig, German) and cultured for 24 hours in nutrient broth at 28°C. A droplet of the yeast solution is taken and placed on the glass substrate. The sample is ready for imaging after oil immersion and without using any coverslip. The low autofluorescence immersion oil (IMMOIL-F30CC, Olympus) is chosen to minimize the autofluorescence background at UV excitation.”

Reviewer #2

1. Please provide detailed information on labelling the sector star target (Fig. 2) and yeast cells (Fig. 5) preparation in the Methods section. Please also comment on photobleaching effects.

Response: we have added detail of sample preparation in method section. The photobleaching has been commented when replying to the point C of the first reviewer.

Line 354 “The metal sector star target is processed with E-beam lithography, E-beam physical vapor deposition, and lift-off. To label the metal sector star target, a mixture solution of dye is extracted from green highlighters (STABILO, German). After dilution in deionized water in a ratio of 1:3, the dye solution is dropped on the metal sector star target and a thin layer forms after the water evaporates.”

Line 349 “Sample preparation: The yeast strain is ordered from DSMZ (Braunschweig, German) and cultured for 24 hours in nutrient broth at 28°C. A droplet of the yeast solution is taken and placed on the glass substrate. The sample is ready for imaging after oil immersion and

without using any coverslip. The low autofluorescence immersion oil (IMMOIL-F30CC, Olympus) is chosen to minimize the autofluorescence background at UV excitation.

Line 279 “the intensity of the NADH autofluorescence is decreased by a factor 2 after 4 minutes of continuous 30 W/cm² UV excitation.”

2. Why is Wiener-filtering applied to the wide-field as well as super-resolved images? In general, Wiener-filtering removes noise but also lowers the optical resolution. Since in the manuscript the resolution of wide-field images is determined or a comparison of wide-field to super-resolution images is performed I would strongly prefer unprocessed (“raw”) data images and analyses performed on these data sets.

Response: A numerical filter, such as the Wiener filter, is necessary for reconstructing the SIM images considering that a deconvolution with the OTF is involved, the experimental data are subject to the presence of noise and the OTF strongly decrease at high spatial frequencies. We applied the Wiener filtering also for the wide field images to highlight only the noise-free super-resolution properties of the SIM technique. It is however true that noise is always present in experimental images and that standard wide-field imaging does not intrinsically require any filtering. Following the reviewer’s advice, we have consequently replaced all the Wiener filtered WF images with unprocessed WF images and analyzed the data sets in the manuscript as reviewer’s suggestion. Note that we have observed no impact of the Wiener filtering on the optical resolution of the raw WF images with the current signal to noise ratio.

Fig.3, fig.4, fig.5, fig. S3 and fig. S4 have been changed and updated with the unprocessed wide-field (WF) images.

3. In order to determine the optical resolution, the authors should use a standardized method, such as Fourier ring correlation.

Response: We sincerely thank reviewer for bringing the Fourier ring correlation (FRC) method to our attention and for recommending this method to determine the optical resolution. We have implemented this approach to confirm the actual optical resolution and attached the results in the main text and the supplementary document.

Line 289 “Using the Fourier ring correlation[34] method, see supplementary section, the wide-field and SIM resolutions are 262 nm and 166 nm respectively.”

See supplementary note 3 on page 4: “The raw and filtered WF images are compared in fig. S2(d) and (e), respectively. The Wiener filter removes the noise outside the bandwidth of the OTF and enhances the contrast of the high-frequency signal but has no impact on the optical resolution limit here. The Fourier ring correlation (FRC) method [1] that is a standardized

approach in the field of microscopy is implemented to confirm the resolution limit of WF and SIM images. The FRC curves of WF and SIM images without applying deconvolution are plotted in Fig. S3. The optical resolutions that are extracted by the intersections between the FRC curves with a conventional threshold of 1/7 are 298 nm and 184 nm for WF and SIM, respectively. The results are in line with the resolutions determined by the FFT peak analysis in Fig. 3(f), namely resolutions of 305 nm and 192 nm, respectively, where a 3 σ criterium has been used to define the threshold."

[34] S. Koho, G. Tortarolo, M. Castello, T. Deguchi, A. Diaspro, G. Vicidomini, "Fourier ring correlation simplifies image restoration in fluorescence microscopy", Nature Communications **10**, 3103 (2019).

REVIEWERS' COMMENTS

Reviewer #1 (Remarks to the Author):

In the revised version, the authors have adequately answered all queries raised during the peer-review process. Furthermore, authors have made the raw files available which will be beneficial for the researchers.

I recommend the publication of this article in Nature Communication .

Reviewer #2 (Remarks to the Author):

The revision has significantly improved the manuscript, and the authors have satisfactorily addressed my questions and concerns. Hence, I would like to recommend the manuscript for publication.